# A Newly Engineered A549 Cell Line Expressing ACE2 and TMPRSS2 Is Highly Permissive to SARS-CoV-2, Including the Delta and Omicron Variants

**DOI:** 10.3390/v14071369

**Published:** 2022-06-23

**Authors:** Ching-Wen Chang, Krishna Mohan Parsi, Mohan Somasundaran, Emma Vanderleeden, Ping Liu, John Cruz, Alyssa Cousineau, Robert W. Finberg, Evelyn A. Kurt-Jones

**Affiliations:** 1Department of Medicine, University of Massachusetts Chan Medical School, Worcester, MA 01655, USA; emma.vanderleeden@umassmed.edu (E.V.); ping.liu@umassmed.edu (P.L.); robert.finberg@umassmed.edu (R.W.F.); evelyn.kurt-jones@umassmed.edu (E.A.K.-J.); 2Diabetes Center of Excellence and Program in Molecular Medicine, University of Massachusetts Chan Medical School, Worcester, MA 01655, USA; krishnamohan.parsi@umassmed.edu (K.M.P.); cousineau95@gmail.com (A.C.); 3Department of Biochemistry and Molecular Biotechnology, University of Massachusetts Chan Medical School, Worcester, MA 01655, USA; mohan.somasundaran@umassmed.edu; 4Department of Pathology, University of Massachusetts Chan Medical School, Worcester, MA 01655, USA; john.cruz@umassmed.edu

**Keywords:** SARS-CoV-2, A549, ACE2, TMPRSS2, EIDD-1931, nirmatrelvir, remdesivir, delta and omicron variants

## Abstract

New variants of severe acute respiratory syndrome coronavirus 2 (SARS-CoV-2) continue to emerge, causing surges, breakthrough infections, and devastating losses—underscoring the importance of identifying SARS-CoV-2 antivirals. A simple, accessible human cell culture model permissive to SARS-CoV-2 variants is critical for identifying and assessing antivirals in a high-throughput manner. Although human alveolar A549 cells are a valuable model for studying respiratory virus infections, they lack two essential host factors for SARS-CoV-2 infection: angiotensin-converting enzyme 2 (ACE2) and transmembrane serine protease 2 (TMPRSS2). SARS-CoV-2 uses the ACE2 receptor for viral entry and TMPRSS2 to prime the SARS-CoV-2 spike protein, both of which are negligibly expressed in A549 cells. Here, we report the generation of a suitable human cell line for SARS-CoV-2 studies by transducing human *ACE2* and *TMPRSS2* into A549 cells. We show that subclones highly expressing ACE2 and TMPRSS2 (“ACE2plus” and the subclone “ACE2plusC3”) are susceptible to infection with SARS-CoV-2, including the delta and omicron variants. These subclones express more ACE2 and TMPRSS2 transcripts than existing commercial A549 cells engineered to express ACE2 and TMPRSS2. Additionally, the antiviral drugs EIDD-1931, remdesivir, nirmatrelvir, and nelfinavir strongly inhibit SARS-CoV-2 variants in our infection model. Our data show that ACE2plusC3 cells are highly permissive to SARS-CoV-2 infection and can be used to identify anti-SARS-CoV-2 drugs.

## 1. Introduction

The COVID-19 pandemic, caused by the SARS-CoV-2 virus, continues to be a global problem, with new variants emerging and breakthrough infections occurring in fully vaccinated individuals. SARS-CoV-2 has infected more than 530 million people and caused over 6.3 million deaths to date (9 June 2022) since its first outbreak in December 2019, according to the WHO Coronavirus (COVID-19) Dashboard [1]. SARS-CoV-2 can infect numerous tissue types in the human body and is particularly destructive to lung epithelial cells. Infection can also result in the deregulation of the immune system, leading to acute respiratory distress syndrome and multi-organ failure in severe COVID-19 patients [2]. In addition, SARS-CoV-2 is highly contagious and highly transmissible (compared with other SARS viruses), and infectious variants are continuously evolving [3].

The infectivity of SARS-CoV-2 strongly depends on the host factor, angiotensin-converting enzyme 2 (ACE2). The SARS-CoV-2 spike (S) glycoprotein interacts with the host ACE2 receptor to initiate S1–S2 cleavage by transmembrane serine protease 2 (TMPRSS2). Furthermore, endosomal proteases release the viral RNA genome into the host cell [4]. Thus, the internalized viral RNA utilizes the host’s cellular protein machinery to counter antiviral responses in the host cell [4]. Developing a robust in vitro system for studying SARS-CoV-2 requires a cell line that accurately models these events.

Although human stem cell-based lung epithelial models are available for studying SARS-CoV-2, they require advanced expertise in cell culture and are costly to propagate for large-scale screening applications [5]. Alternative cell lines, such as human lung-derived Calu-3 and colon-derived Caco-2 cells, are permissive to SARS-CoV-2 infection and show virus replication over a period of 120 h. However, neither cell line shows comparable infectivity to Vero E6 cells or displays SARS-CoV-2-induced substantial cell damage and cytopathic syncytia formation [6], emphasizing the need for an improved platform for studying SARS-CoV-2 biology and identifying antivirals.

Lentiviral transduction is an efficient method of expressing transgenes in mammalian cells through stable chromosomal integration; however, gene integration is random and generates a heterogeneous population of transduced cells [7]. In previous studies, we used lentiviral methods to generate A549 cells (43.20) expressing ACE2 and TMPRSS2 [8]. We noticed that the susceptibility of SARS-CoV-2 infection in 43.20 cells was dramatically reduced after several rounds of sequential culture. This observation might be associated with the reduced ACE2 expression level in 43.20 cells. Intriguingly, Sherman et al. recently reported that the expression of ACE2 on the cell surface is heterogeneous in Huh-7 and Calu-3 cell lines [9]. They found that ACE2 expression was unstable after sorting, and the proportion of positive cells gradually reverted to the parental distribution after several passages. In other words, the expansion of ACE2-positive cells and cellular ACE2 surface expression increased after sorting but subsequently decreased over multiple generations of daughter cells. This phenomenon is similar to what we observed in the 43.20 cell model.

Here, we report the development of a robust A549-based cell line that highly expresses ACE2 and TMPRSS2 and remains relatively stable to SARS-CoV-2 infection susceptibility after multiple passages, providing a valuable cell line for performing high-throughput in vitro testing to evaluate the efficacy of SARS-CoV-2 antivirals and facilitate research on drugs for COVID-19 treatment.

## 2. Materials and Methods

### 2.1. Virus, Cells, Plasmids, Antibodies, and Antiviral Compounds

Low-passage SARS-CoV-2 stocks were used in this study. USA-WA1/2020 strain (BEI#NR-52281) was propagated by infecting Vero E6 cells at a low multiplicity of infection (MOI) and then maintaining the infected cells in Eagle’s Minimal Essential Medium (EMEM, Millipore Sigma, Burlington, MA, USA) with 2% fetal bovine serum (FBS, Atlas Biologicals, Fort Collins, CO, USA). Cell-free virus supernatant was harvested 72 h post-infection when < 70% cytopathic effects were evident. Aliquots of concentrated virus stocks were stored at −80 °C, titered by tissue culture infectious dose (TCID_50_), and used to infect the different cell lines at an MOI of 0.1 or 0.4. An mNeonGreen-labeled infectious clone-derived strain (icSARS-CoV-2-mNG) was received from the World Reference Center for Emerging Viruses and Arboviruses at the University of Texas Medical Branch after obtaining a material transfer agreement between the two institutions. The virus was reconstituted and propagated according to the previously described method [10]. Briefly, the lyophilized virus was reconstituted with 0.5 mL of sterile water, and aliquots of 25 µL and 50 µL were stored in a −80 °C freezer. An aliquot of the reconstituted virus was used for a low MOI infection of ~90% confluent Vero E6 cells. The infected cells were maintained in EMEM with 2% FBS, and the cell-free virus supernatant was harvested, stored, and titered as described for the WT virus. The two variants, B.1.617.2 delta (BEI#NR-55672) and B.1.1.529 omicron (BEI#NR-56461), were propagated by infecting Vero E6 TMPRSS2-T2A-ACE2 cells at a low MOI 0.01 and maintained in EMEM with 2% FBS. The cell-free virus supernatant was harvested, stored, and titered as described for the WT virus. All work with infectious SARS-CoV-2 was performed in a biosafety level 3 laboratory facility (BSL-3) by personnel trained to handle BSL-3 agents according to standard operating procedures and approved by the University of Massachusetts Chan Medical School Institutional Biosafety Committee.

A549 (CCL-185), Calu-3 (HTB-55), and Vero E6 (CRL-1586) cells were obtained from ATCC. Vero E6-TMPRSS2-T2A-ACE2 (NR-54970) cells were provided by BEI Resources. Commercial cell lines A549-hACE2 (a549-hace2, lot 42-01) and A549-hACE2-TMPRSS2 (a549-hace2tpsa, lot 42-02) were purchased from InvivoGen in February 2021. A549-43.20 cells were generated by Dr. Maehr’s laboratory as described in Koupenova et al. [8]. Cells were cultured at 37 °C in Dulbecco’s Modified Eagle Medium (DMEM) supplemented with 10% heat-inactivated FBS, 100 U/mL penicillin–streptomycin, and 1% L-glutamine. The ACE2plusC3 cell line is available from the ATCC (CRL-3560).

The plasmid sets for generating the SARS-CoV-2 S pseudotyped virus were provided by BEI Resources (NR-52948; NR-53765). The generation of pseudotyped lentiviral particles was based on the previously described protocol [11]. The pHAGE-EF1a-ACE2-PGK-puroR and pHAGE-EF1a-TMPRSS2-PGK-puroR plasmids were used to establish A549-43.20. Anti-ACE2 and anti-TMPRSS2 antibodies were purchased from R&D Systems (Minneapolis, MN, USA, Catalog No. AF933) and Santa Cruz Biotechnology, Inc. (Dallas, TX, USA, Catalog No. sc-515727), respectively.

Decanoyl-RVKR-CMK (#3501) was purchased from TOCRIS (Bristol, UK). Camostat mesylate (SML0057) and naphthofluorescein (#70420) were purchased from Millipore Sigma (Burlington, MA, USA). Nirmatrelvir (HY-138687), EIDD-1931 (HY-125033), remdesivir (HY-104077), nelfinavir mesylate (HY-15287A), E64d (HY-100229), and fluvoxamine (HY-B0103) were purchased from MedChemExpress (Monmouth Junction, NJ, USA). Dimethyl sulfoxide (DMSO) was purchased from Sigma-Aldrich (St. Louis, MO, USA, Catalog No. D8418).

### 2.2. RNA Isolation and RT-PCR

RNA was isolated from virus-containing medium or cell lysates using TRIzol LS^TM^ and stored at −80 °C for RT-qPCR. The expression level of isolated viral RNA was quantified using the QuantiFast Pathogen RT-PCR Kit (Qiagen, Germantown, MD, USA, Catalog No. 211352) and the 2019-nCoV RUO Kit (IDT, Redwood City, CA, USA, Catalog No. 10006713). The cycling conditions followed the manufacturer’s protocol. Isolated cellular RNA was incubated in gDNA Wipeout Buffer to remove contaminating genomic DNA. The purified RNA was then used for cDNA synthesis using the QuantiTect Reverse Transcription Kit (Qiagen, 205311). The resultant cDNA was used to measure mRNA expression levels of *ACE2* and *TMPRSS2* by qPCR with gene-specific primers (human *ACE2*, sense 5′-GGGATCAGAGATCGGAAGAAGAAA-3′, and antisense 5′-AGGAGGTCTGAACATCATCAGTG-3′; human *TMPRSS2*, sense 5′-AATCGGTGTGTTCGCCTCTAC-3′, and antisense 5′-CGTAGTTCTCGTTCCAGTCGT-3′), and Applied Biosystems^TM^ SYBR Green reagent (Thermo Fisher Scientific, Waltham, MA, USA, Catalog No. 4309155). *GAPDH* was used as an endogenous control gene (sense 5′-TCCTCCACCTTTGACGCT-3′ and antisense 5′-TCTTCCTCTTGTGCTCTTGC-3′).

### 2.3. Plaque Assay

Approximately 2 × 10^5^ Vero E6 cells were seeded in each well of 12-well plates and grown at 37 °C in 5% CO_2_ for 18 h. The virus was serially diluted in 1× Minimum Essential Media (MEM) with 3% FBS. Cell monolayers were aspirated and inoculated with 300 μL of virus inoculum. The infected cells were incubated at 37 °C with 5% CO_2_ for 1 h. The virus-containing medium was then aspirated from the cells and replaced with an overlay medium containing 1× MEM with 0.42% bovine serum albumin (BSA), 20 mM HEPES, 0.24% NaHCO_3_, and 0.7% agarose (Thermo Fisher Scientific, Waltham, MA, Catalog No. LP0028). After a 72-h incubation, the cells were fixed with 4% paraformaldehyde (PFA) overnight and then stained with crystal violet solution (Sigma-Aldrich, St. Louis, MO, USA) and quantified.

### 2.4. Immunofluorescence Staining, Cell Sorting, and Flow Cytometry

Cells were plated on 96-well tissue culture plates (Corning Inc., Corning, NY, USA) and infected with low-passage SARS-CoV-2 or icSARS-CoV-2-mNG at the indicated MOIs and incubated for designated times. Infected cells were fixed with 4% PFA for 30 min at room temperature, gently washed twice with phosphate-buffered solution (PBS), and then permeabilized with 1% Triton X-100 in PBS. The cells were then blocked with 5% BSA. The fixed cells were treated with either a human monoclonal primary antibody conjugated with Alexa-488 targeting the SARS-CoV-2 S antigen at 1:200 dilution [12] or a mouse monoclonal primary antibody targeting the nucleocapsid protein (NP) antigen (SinoBiological, Beijing, China, Catalog No. 40143-MM08) at 1:1000 dilution and incubated for 2 h at 4 °C. After washing the cells with 0.05% Tween 20 solution, they were treated with an anti-mouse goat secondary antibody conjugated with Alexa-594 (Invitrogen, Carlsbad, CA, USA, Catalog No. A-11005) at 1:2000 dilution and incubated for 1 h at 4 °C. The cells were then counterstained with 4′,6-diamidino-2-phenylindole (DAPI; Abcam, Cambridge, MA, USA) for 15 min at 4 °C to visualize the cell nuclei. Images were acquired with Celigo Image Cytometer or ImageXpress Micro-XL (IXM) system by immunofluorescence with 4× and 10× objectives. The images were processed by Celigo Image Software (Nexcelom Bioscience, Lawrence, MA, USA, version 200-BFFL-5C) or Meta-Xpress Software (Molecular Devices, San Jose, CA, USA, version 4.0.0.43).

To increase the infectivity of the cells, the ACE2+ populations were further sorted using an ACE2-specific antibody to establish an ACE2plus cell line and its subclones. Briefly, trypsinized cells were stained with goat anti-ACE2 antibody (R&D, AF933) and sorted using a BD FACSII sorter (BD Biosciences, Franklin Lake, NJ, USA), located in the Flow Cytometry Core, UMass Chan Medical School. For flow cytometric analysis, freshly trypsinized cells were stained with goat anti-ACE2 antibody (R&D, AF933) on ice for 30 min. After wash, cells were incubated with Alexa-647-conjugated donkey anti-goat IgG antibody (A-21447, Invitrogen) on ice for a half hour, then washed and resuspended in an appropriate volume of cell-sorting buffer (1× PBS, 0.5% BSA, 2 mM EDTA). Flow cytometric analysis was performed using a BD LSRII Flow Cytometer.

### 2.5. Luciferase Assay and Cytotoxicity

SARS-CoV-2 S pseudotyped virus activity was determined using the Bright-Glo Luciferase Assay System (Promega, Madison, WI, USA). Luminescence was quantified at 48 h post-virus (or mock) infection of the ACE2plus cells. Cell death was measured using the LDH Cytotoxicity Detection Kit from Roche (11644793001) to measure the lactate dehydrogenase activity in the supernatants.

### 2.6. Statistical Analyses

Data are expressed as means ± standard deviations (SD), and the significance of differences between groups was evaluated using ANOVA with Dunnett’s multiple comparisons test. All tests were performed using Prism 9 (GraphPad Software, v9).

## 3. Results

### 3.1. Establishing an ACE2plus Cell Line That Is Highly Permissive to SARS-CoV-2 Infection

To establish a robust and stable human lung alveolar cell line for SARS-CoV-2 studies, we transduced A549 cells with human ACE2-expressing lentivirus at a low MOI and selected them with puromycin (1 μg/mL). Puromycin-resistant cells were further transduced with human TMPRSS2-expressing lentivirus at a high MOI as described in [8] and shown in Figure 1A. Following limiting-dilution cloning, over 50 clones were selected and challenged with SARS-CoV-2 USA-WA1/2020 (referred to as “wild type”, WT). Of the 50+ clones, clone 43.20 exhibited the most efficient SARS-CoV-2 infection rate, which was further enriched by cell sorting with an ACE2-targeting antibody as described in Materials and Methods. Following another round of puromycin selection (5 μg/mL), the infectivity significantly increased to ~40% (Figure 1B,C). Next, we compared the infectivity of ACE2plus and Vero E6 cells. Each cell type was challenged with WT SARS-CoV-2 or the mNeonGreen-labeled SARS-CoV-2 strain icSARS-CoV-2mNG (WT-mNG) at MOIs of 0.05 and 0.1 for 48 h (control cells received growth medium alone). The ACE2plus cells exhibited infection rates of ~60%, similar to the infection rate in the Vero E6 cells (~70%) (Appendix A). The two lines showed similar viral nucleoprotein transcript levels in the supernatants (Appendix A).

Next, we conducted single-cell sorting of ACE2plus cells to generate single-cell-derived ACE2plus subclones (Figure 2A). Over 20 clones were isolated and challenged with WT SARS-CoV-2, with most clones showing >60% infectivity. To test its susceptibility to different SARS-CoV-2 variants, we infected ACE2plusC3 cells with the WT virus or the delta and omicron variants and compared them with commercial A549-hACE2-TMPRSS2 cells (“IVG-AT”) used by researchers studying SARS-CoV-2 [13,14,15]. As shown in Figure 2B, viral nucleocapsid protein (NP) was detected in most ACE2plusC3 cells 48 h post-infection. However, NP signals were sparse in IVG-AT cells, and we rarely observed virus-induced cell fusion. Furthermore, ACE2plusC3 cells showed superior infectivity with WT SARS-CoV-2 and the delta and omicron variants (Figure 2C). Altogether, these results demonstrate that ACE2plusC3 cells are highly susceptible to SARS-CoV-2 infection.

### 3.2. Characterization of the ACE2plus and ACE2plusC3 Cell Lines

ACE2 and TMPRSS2 are key receptors for SARS-CoV-2 entry. To measure levels of *ACE2* and *TMPRSS2* mRNA expression, RT-qPCR was performed on cell lysates collected from ACE2plus, ACE2plusC3, parental A549, Calu-3, and two commercial cell lines—A549-hACE2 (“IVG-A”, lot 42-01) and A549-hACE2-TMPRSS2 (“IVG-AT”, lot 42-02) (Figure 3A,D). As expected, both *ACE2* and *TMPRSS2* mRNA expression levels in the parental A549 cells were negligible. *ACE2* mRNA expression levels in ACE2plus and IVG-AT cells were similar. However, ACE2plus cells expressed a higher level of *TMPRSS2* mRNA than IVG-AT and Calu-3 lines. Next, we conducted flow cytometry to determine the cell-surface ACE2 protein expression. Over 95% of ACE2plus cells were ACE2-positive, whereas IVG-AT cells were only 33% ACE2-positive (Figure 3B). IVG-AT cells grow slowly, with doubling times of approximately 50 h (Figure 3C). In contrast, the growth rate of ACE2plus cells is faster and more consistent than the commercial cell line.

We then examined ACE2 and TMPRSS2 expression in the ACE2plusC3 cell line and determined that these cells expressed higher mRNA levels of *ACE2* and *TMPRSS2* than the parental ACE2plus cells (Figure 3D). We confirmed this finding using co-immunofluorescence staining and immunoblotting (Figure 3E,F). A549 cells were included as a control. As expected, the ACE2plusC3 cells showed strong and ubiquitous expression for both proteins, which may contribute to the stable virus infectivity between the early and late passages (Figure 3G). According to recent reports, the spike D614G mutation is associated with ACE2 receptor binding and virus entry [16]. To test this, we used the 614D and 614G strains of S-pseudotyped lentiviral particles to infect ACE2plusC3 cells. The spike 614G virus showed stronger infectivity than 614D in a dose-dependent manner (Appendix A). These data support the use of ACE2plusC3 for monitoring SARS-CoV-2 pseudotyped lentiviral infection. Our data are also in line with those reported by Cheng et al. (2021), showing that D614G substitution increased the virus titer. Taken together, these results provide evidence that the ACE2plusC3 cell line is a suitable model for SARS-CoV-2 research.

### 3.3. Determining If ACE2plusC3 Cells Can Be Used to Test the Efficacy of Antiviral Drug Candidates against WT SARS-CoV-2 Infection

To test the utility of the ACE2plusC3 model in antiviral drug screening, we used these cells to compare the antiviral efficacy of nine previously identified or potential antiviral drug candidates against SARS-CoV-2 infection. Nirmatrelvir, EIDD-1931, and remdesivir have been approved by the FDA to treat COVID-19. Nirmatrelvir (PF-07321332) is an orally bioavailable viral 3C-like protease inhibitor [17]. EIDD-1931, the active metabolite of EIDD-2801, is an orally bioavailable drug that targets viral RNA-dependent RNA polymerase [18]. Nelfinavir, a leading HIV protease inhibitor, has been shown to target the SARS-CoV-2 main protease to inhibit virus replication [19]. Camostat mesylate, naphthofluorescein, E64d, and decanoyl-RVKR-CMK have been widely used to study the virus entry and spike processing by targeting host cell proteases [20]. Fluvoxamine is a selective serotonin reuptake inhibitor that is approved by the FDA to treat obsessive–compulsive disorder. This drug binds to the sigma-1 receptor on immune cells to reduce inflammatory cytokine production [21] and is being evaluated for the treatment of COVID-19 in randomized controlled trials in humans [22].

To determine the dose–response curves (DRC) of the antivirals, we inoculated ACE2plusC3 cells with WT SARS-CoV-2 for 1 h and incubated them with each drug candidate for 48 h before fixation. Viral nucleocapsid protein (NP) was stained using immunofluorescence, and cell nuclei were stained with DAPI; fluorescence was quantified to determine the inhibition efficacy using the IXM image system and software. The DRC analysis of the reference drugs (i.e., EIDD-1931, remdesivir, and nirmatrelvir) showed strong inhibition efficacy against SARS-CoV-2-infected ACE2plusC3 cells (Figure 4). Nelfinavir also showed a potent and dose-dependent inhibition of infection, but we noted increased toxicity with ≥25 μM. Next, we treated cells with camostat mesylate, naphthofluorescein, E64d, or fluvoxamine. Only camostat mesylate treatment significantly inhibited SARS-CoV-2 infection without causing cell toxicity (Figure 4). This is consistent with recent reports that camostat mesylate significantly reduces SARS-CoV-2-driven entry and infection in primary human lung cells [4]. Since several published studies have reported that the cleavage of the SARS-CoV-2 S protein at a putative furin cleavage site (RRARS) at R685/S686 is critical for virus entry [16,23,24], we investigated the efficacy of the furin inhibitor decanoyl-RVKR-CMK and observed a moderate inhibition of infection in the WT SARS-CoV-2-infected cells as determined by immunofluorescence and plaque assays (Appendix A). These results demonstrate that our ACE2plucC3 cell model can be used to evaluate antiviral drugs and potentially developed for high-throughput screening.

### 3.4. Assessing the Utility of ACE2plusC3 Cells to Identify Potent Antivirals against Emerging Variants of SARS-CoV-2

Given that the emergence of SARS-CoV-2 variants with increased transmissibility continues to threaten global public health, we tested the efficacy of a panel of antivirals at inhibiting the delta and omicron variants of SARS-CoV-2 in our ACE2plusC3 cells. We found that EIDD-1931, remdesivir, nirmatrelvir, and nelfinavir exhibit a dose-dependent inhibition of infection. EIDD-1931 reduces WT, delta, and omicron SARS-CoV-2 infection by at least 50% at 1 μM (Figure 5 top), with complete inhibition at 5 μM. Remdesivir inhibited the WT virus and the two variants at 1 μM. A very low concentration of nirmatrelvir (0.1 μM) reduced delta virus infection by 50% and nearly completely inhibited WT and omicron SARS-CoV-2 infection. By contrast, nelfinavir did not completely inhibit infection at a higher concentration of 10 μM (Figure 5 bottom). Compared with WT and delta SARS-CoV-2, the omicron variant appears more sensitive to the drug treatment in this model.

## 4. Discussion

The world is now in the third year of the SARS-CoV-2 pandemic and facing new challenges from emerging variants. While novel vaccines provide protection and slow the spread of infection, this approach is not feasible in some immunocompromised individuals, and breakthrough infections in vaccinated and boosted individuals have been reported [25]. Antiviral strategies are a promising alternative; thus, our aim was to establish a reliable cell culture system for accelerating the discovery of novel antiviral drugs against SARS-CoV-2 and its variants. We generated the ACE2plusC3 cell line that allows researchers to quantitate SARS-CoV-2 infectivity through high-throughput screening of small molecules. Common assays using the pseudotyped virus for antiviral research can be easily transferred to the ACE2plusC3 setting with comparable results.

Several cell culture models are available for monitoring SARS-CoV-2 infection and pseudotyped virus activity. Although 293T-ACE2 and Vero E6 have been widely used to study SARS-CoV-2 entry, replication, and antiviral treatments [3,4,23], these cells are unsuitable for investigating the pathological mechanisms of the host cell’s response to virus infection as they are not derived from human lung tissue, cannot be used to assess cytopathic effects, and do not express type I interferon genes [26]. Calu-3 cells derived from human lung tissue are an alternative system; however, these cells grow slowly and their ACE2 surface abundance is heterogeneous [9,27]. Unlike Calu-3 cells, A549 cells grow faster and are easier to handle in cell culture [27]. Here, we compared the properties of the commercialized A549-hACE2/TMPRSS2 line (“IVG-AT”) with single-cell-derived ACE2plusC3 cells and found that the ACE2plusC3 cells offer significant advantages in terms of cell proliferation and SARS-CoV-2 susceptibility. In addition, ACE2plusC3 cells can be subjected to more antibiotic selection than IVG-AT to establish required stable cell lines. Another advantage is that ACE2plusC3 cells are a more homogeneous cell population than either the parental ACE2plus cell line or the commercial IVG-AT cell line. Our cell model also displayed more sensitive responses to antiviral treatment when compared with commonly used Vero E6 and Calu-3 cell lines [28,29], demonstrating that ACE2plusC3 cells are a valuable system for assessing antivirals.

Antivirals can target multiple steps in the SARS-CoV-2 entry process, including virus attachment to the cell surface, receptor engagement, protease processing, and membrane fusion [30]. The entry process mainly relies on the receptor-binding domain (RBD) and S2 domain in the SARS-CoV-2 spike protein. Viral entry also depends on host factors. In addition to ACE2, several molecules have been suggested to serve as alternative receptors for SARS-CoV and SARS-CoV-2 entry, including C-type lectins, DC-SIGN, and L-SIGN [31]. These receptors bind a wide range of viruses by recognizing the glycans on the virion surface and promote viral entry by allowing the virus to attach to the target cell, acting as attachment factors for virus entry. Although those receptors increase virus entry, they do not support virus infection in the absence of the ACE2 receptor [32,33]. In this work, we show that ACE2pluC3 cells were susceptible to infection by multiple SARS-CoV-2 variants, allowing for studies with a broad range of viral strains. Using ACE2plusC3 cells, researchers can now examine additional attachment factors and their contributions to susceptibility to virus infection.

Our results demonstrate the potential of ACE2plusC3 cells for examining SARS-CoV-2 variants, particularly the omicron variant that has been reported to have unique properties [34]. In our system, the omicron variant demonstrated decreased infectivity compared with the WT virus and the delta variant. This decrease could be due to attenuation. Although we used a low-passage virus stock to mitigate attenuation, further studies are needed to rule it out. In our antiviral drug test, we also found that the omicron variant appears more sensitive to antiviral drug treatment than the WT and delta variant. Recently, Hui et al. reported that omicron (B.1.1.529) replicates faster than other SARS-CoV-2 variants in the human bronchi but less efficiently in the lung parenchyma [35]. They found that omicron infection was less dependent on TMPRSS2 activities and could enter cells primarily through the endocytic pathway, while delta preferentially enters cells through cell surface fusion. It is unclear whether omicron is more dependent on the endocytic pathway for entering cells or if the omicron spike protein inefficiently uses the cellular protease TMPRSS2 for virus entry in our cell model. The omicron variant has 30 mutations in the spike protein, half of which are in the RBD, suggesting that this variant may be immunologically resistant to antibody-mediated protection [36]—further emphasizing the need for antivirals. Moreover, the spike protein is just one of the structural proteins. Other viral proteins, like the membrane (M), envelope (E), and nucleocapsid (N) proteins, may also contribute to virus entry and attenuate replication. Interestingly, Gerard Goh et al. suggested that the omicron virion has a harder shell with attenuated replication than others based on their computed data from a shell disorder model [37]. According to their prediction, the outer shell disorder of omicron is lower than that of other variants and might provide the omicron virion with a rigid outer shell that protects the virus from damage by salivary or mucosal antimicrobial enzymes. In a similar strategy, they predicted that the inner shell disorder of the omicron is inherently attenuated, like pangolin-CoV 2017, an attenuated precursor of SARS-CoV-2 that might have jumped from pangolins to humans in 2017. Using the ACE2plusC3 system, we can begin to address these questions about the omicron variant and future variants.

In conclusion, we developed a suitable human cell model for SARS-CoV-2 susceptibility. Our data on authentic virus infection, pseudotyped virus infection, and antiviral assays highlight the potential of the ACE2plusC3 cell model for studying emerging SARS-CoV-2 variants and antiviral drug screens.

## Figures and Tables

**Figure 1 viruses-14-01369-f001:**
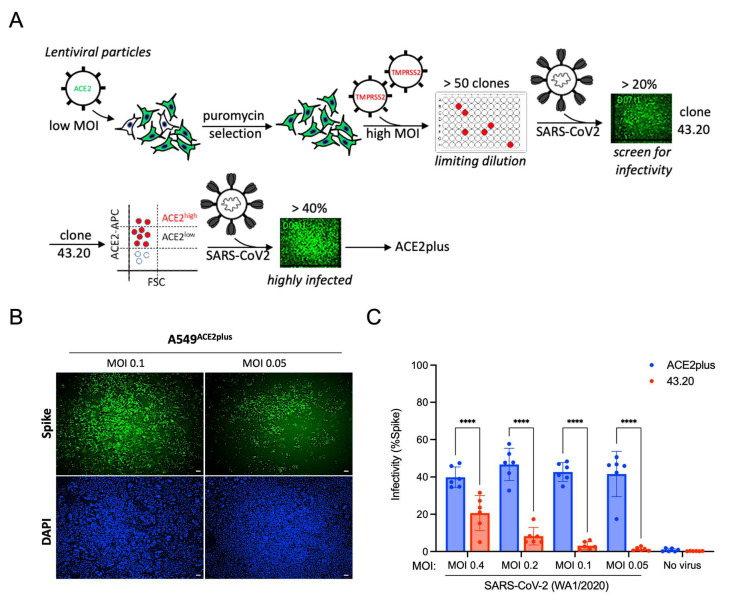
Establishing a highly permissive ACE2plus cell line for SARS-CoV-2 infection. (**A**) Experimental scheme to establish A549-ACE2-TMPRSS2 (43.20) and A549-ACE2-TMPRSS2 (ACE2plus) cells. (**B**) Immunofluorescence staining of the spike protein of SARS-CoV-2 at 48 h post-infection. Scale bar, 100 μm. (**C**) ACE2plus cells show increased infectivity. The percentage of infected cells was quantified using ImageXpress to determine the infectivity (spike + cells in total cell number). Data are expressed as mean ± SD. *n* = 6 biological replicates. ****, *p* < 0.0001.

**Figure 2 viruses-14-01369-f002:**
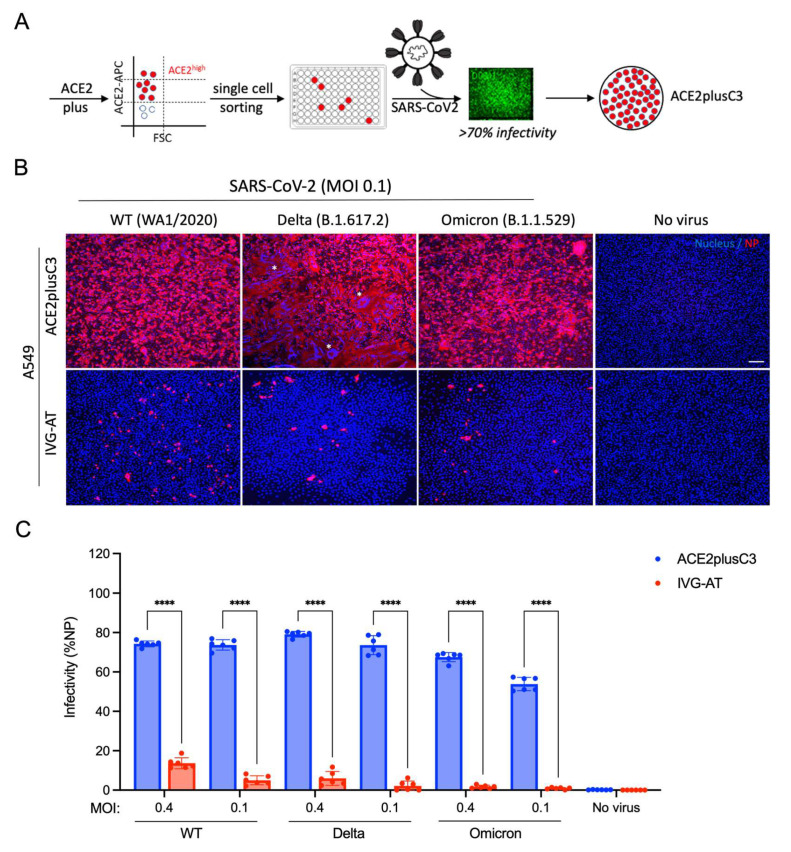
The characterization of single-cell clones from the ACE2plus cell population. (**A**) The experimental scheme to establish the ACE2plusC3 line. The commercial A549-hACE2-TMPRSS2 cell line (IVG-AT) was tested using the same experimental conditions. (**B**) Virus-infected cells are visualized by immunofluorescence staining of the nucleocapsid protein of SARS-CoV-2 at 48 h post-infection. Scale bar, 200 μm. Syncytia are indicated by asterisks. (**C**) The percent infectivity is determined by the number of NP+ (nucleocapsid protein) cells out of the total cells. The data are expressed as mean ± SD. *n* = 6 biological replicates. ****, *p* < 0.0001.

**Figure 3 viruses-14-01369-f003:**
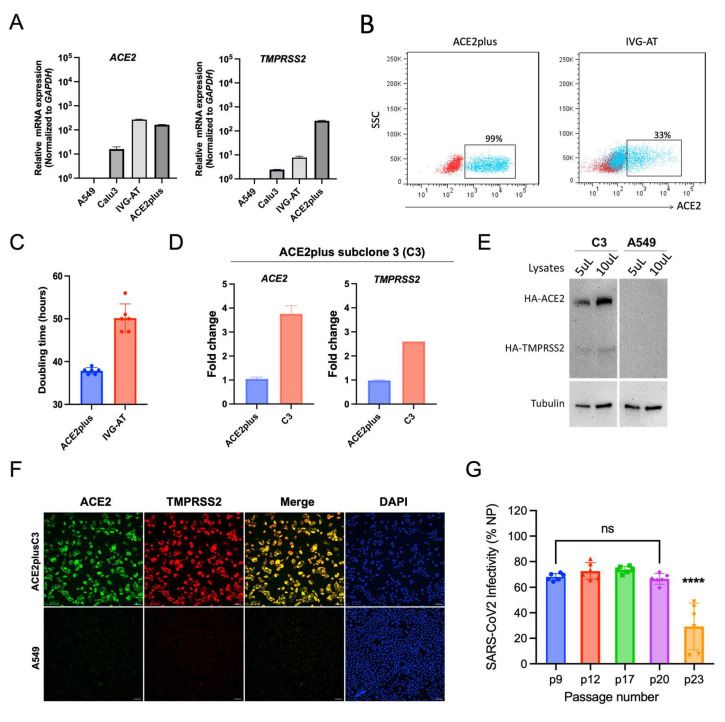
The characterization of the ACE2plus and ACE2plusC3 cell lines. The mRNA expression levels of *ACE2* and *TMPRSS2* in the indicated cell lines were measured with RT-qPCR (**A**,**D**). IVG-A and IVG-A/T commercial cell lines were used as comparators. (**B**) The cell surface ACE2 expression level was measured with flow cytometry using live cells. (**C**) The cell-doubling time was determined using a cell growth curve and Celigo Image software. (**E**) The protein expression levels of ACE2 and TMPRSS2 in cells were examined by immunoblotting and immunofluorescence staining (**F**). Scale bar, 100 μm. (**G**) The virus infectivity remains stable between early and late passages (p9, p12, p17, p20, and p23). ACE2plusC3 cells were infected with WT SARS-CoV-2 at an MOI of 0.1 for 48 h. Data are expressed as mean ± SD. *n* = 6 biological replicates. ns, not significant; ****, *p* < 0.0001.

**Figure 4 viruses-14-01369-f004:**
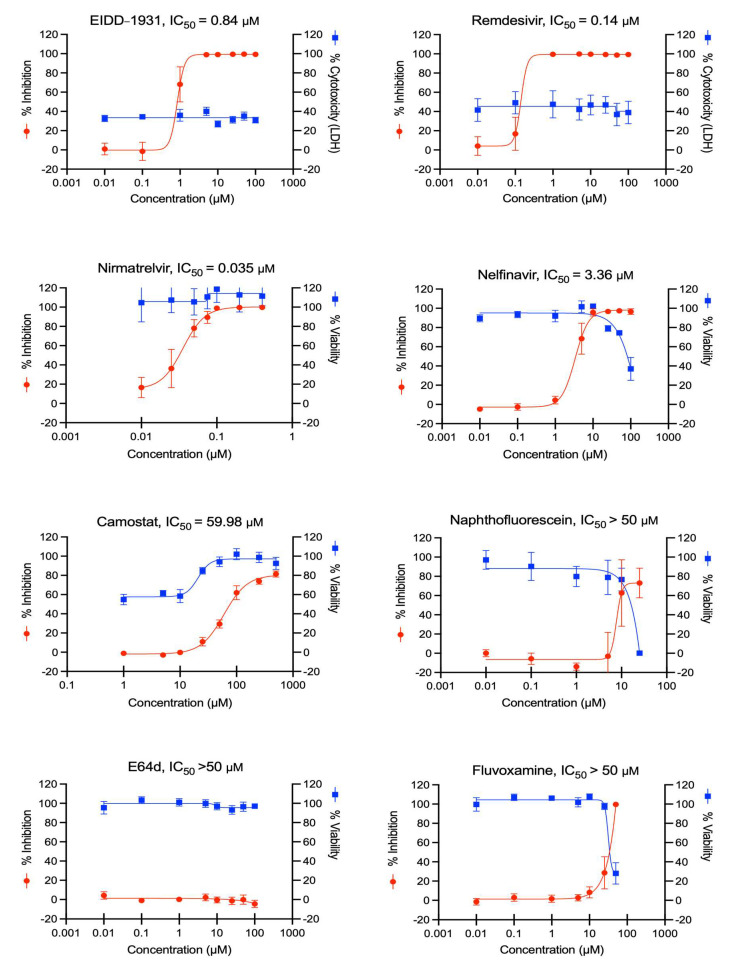
The antiviral activity of EIDD-1931, remdesivir, nirmatrelvir, and other small molecules in ACE2plusC3 cells. Cells were infected with WT SARS-CoV-2 at an MOI of 0.1 for 48 h in the presence of the indicated concentrations (μM). Infection was calculated by dividing the number of infected cells (measured by immunofluorescence of SARS-CoV-2 nucleocapsid protein) by the total cell nuclei present (measured by DAPI). The red line indicates the inhibition of SARS-CoV-2 infection, and the blue line indicates cell viability. Each condition was performed in sextuplicate with averages and standard deviations indicated. The curves were fitted using GraphPad Prism software. Half maximal inhibitory concentration (IC_50_) was calculated from the curve fit. Cell viability was measured by LDH assay and by cell morphology and cell number.

**Figure 5 viruses-14-01369-f005:**
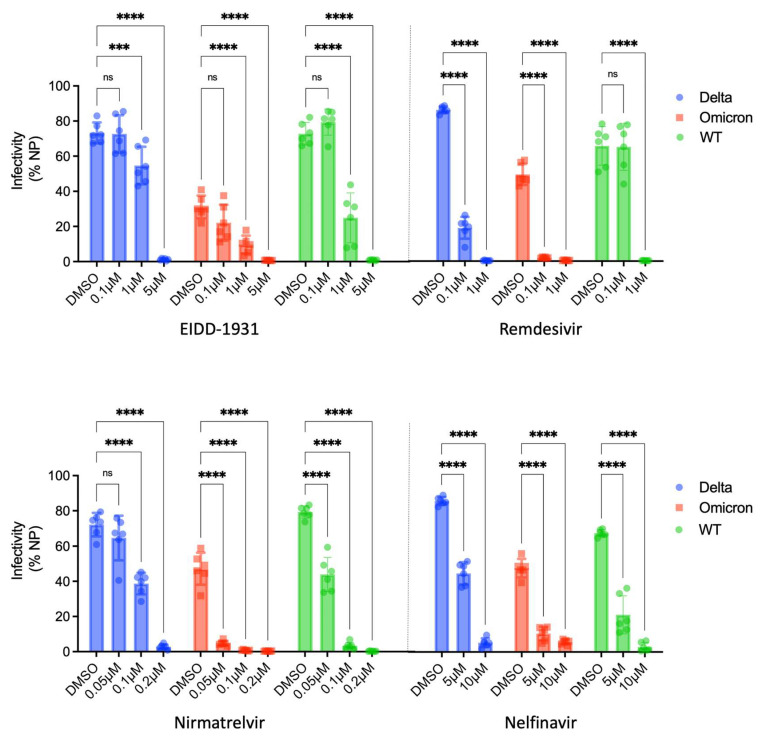
The antiviral activity of EIDD-1931, remdesivir, nirmatrelvir, and nelfinavir against wild-type, delta, and omicron SARS-CoV-2 infection in ACE2plusC3 cells. Small molecules were evaluated in ACE2plusC3 cells at the indicated concentrations (μM) or with DMSO control. Cells were infected with wild-type SARS-CoV-2 and the indicated variants at an MOI of 0.1 for 48 h. Virus-infected cells were visualized with immunofluorescence staining of SARS-CoV-2 NP. Infectivity was measured as described in Figure 1. Each concentration was performed in sextuplicate with averages and standard deviations indicated. ns, not significant; ***, *p* < 0.001; ****, *p* < 0.0001.

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
