# Peer review of "A Newly Engineered A549 Cell Line Expressing ACE2 and TMPRSS2 Is Highly Permissive to SARS-CoV-2, Including the Delta and Omicron Variants"

_viruses, 2022, doi:10.3390/v14071369_

Round 1

Reviewer 1 Report

The manuscript by Chang and colleagues is well written and executed. However, it needs a bit more polish before being accepted for publication. 

Minor Comments : 

Line 44: Mention the exact date 

Line 65 on words: The authors should cite the supporting references of their previous work. 

Line: 73 onwards: Revise the entire section. It is good to raise questions in the introduction; however,  avoid discussing results. 

Line: 134: correct CO2

Figure 1c, Figure 3, and Figure 5: P values missing. 

Author Response

The manuscript by Chang and colleagues is well written and executed. However, it needs a bit more polish before being accepted for publication. 

>We thank the reviewer for the affirmation of the quality of the experiments. We have revised the manuscript and included additional references and a more in-depth discussion of the implications and significance of the work. This revised manuscript has been edited by native English-speaking colleagues.

Minor Comments: 

Line 44: Mention the exact date 
> We have mentioned the exact date, “June 9, 2022,” in line 48.

Line 65 on words: The authors should cite the supporting references of their previous work. 
> We have cited our previous work in line 74, as shown in reference 8: “Koupenova et al., SARS-CoV-2 Initiates Programmed Cell Death in Platelets. Circ Res 2021, 129(6):631-646.”

Line: 73 on wards: Revise the entire section. It is good to raise questions in the introduction; however, avoid discussing results. 
> We thank the reviewer for the constructive comment. We have revised the entire section (lines 70–88).

"Lentiviral transduction is an efficient method of expressing transgenes in mammalian cells through stable chromosomal integration; however, gene integration is random and generates a heterogeneous population of transduced cells [7]. In previous studies, we used lentiviral methods to generate A549 cells (43.20) expressing ACE2 and TMPRSS2 [8]. We noticed that the susceptibility of SARS-CoV-2 infection in 43.20 cells was dramatically reduced after several rounds of sequential culture. This observation might be associated with the reduced ACE2 expression level in 43.20 cells. Intriguingly, Sherman et al. recently reported that the expression of ACE2 on the cell surface is heterogeneous in Huh-7 and Calu-3 cell lines [9]. They found that ACE2 expression was unstable after sorting, and the proportion of positive cells gradually reverted to the parental distribution after several passages. In other words, the expansion of ACE2-positive cells and cellular ACE2 surface expression increased after sorting but subsequently decreased over multiple generations of daughter cells. This phenomenon is similar to what we observed in the 43.20 cell model.

Here, we report the development of a robust A549-based cell line that highly expresses ACE2 and remains relatively stable to susceptibility to SARS-CoV-2 infection after multiple passages, providing a valuable cell line to perform high-throughput in vitro testing to evaluate the efficacy of SARS-CoV-2 antivirals and facilitate research on drugs for COVID-19 treatment."

Line: 134: correct CO2
> We have corrected CO2 to “CO2 ” in line 154

Figure 1c, Figure 3, and Figure 5: P values missing. 
> We have added “p < 0.0001” in Figure 1 (line 227),  “ p < 0.01 and p < 0.0001” in Figure 3 (line 278), and “p < 0.0001” in Figure 5 (line 365).

Reviewer 2 Report

This is a reasonably good experimental paper.

  I am, however, somewhat skeptical of the implications of the results.put forth in this  paper.    Part of the problem has to do with the tendency of current prevailing research  to overemphasize on the S protein and ACE2 while being totally oblivious of the fact that  there are other SARS-CoV-2 proteins.  This is evident by the observation that much of  the current COVID-19 protein research involve the S protein but very few involve  other SARS-CoV-2 proteins. I think this constitutes a fundamental flaw in existing research as, while the S protein is an important protein especially in viral entry and antibody recognition, it is not necessarily the most important and certainly, not the only SARS-CoV-2 protein.   While, I think the paper is fundamentally correct, I suspect that there is more to it than what the authors believe they have found. The authors should not have simply assumed  their hypothesis was correct without doing a literature search for alternative possibilities involving other proteins.   1) The basic hypothesis that is being addressed in this paper at tributes the reduced ability of the virus to infect A549 cells that were sequentially cultured to the "cell growth 

heterogeneity in ACE2 and TMPRSS2 expressing cells".  While It is not

clear how the virus was prepared, it possible that the virus had been attenuated by the several passage of the virus through various cell lines. In fact, this is thr classical  method used to attenuate poliovirus. The poliovirus is passed through many generations of cell-line before it becomes attenuated: 
https://www.cdc.gov/vaccines/pubs/pinkbook/polio.html   My question is then: Is it possible that what the authors saw was the attenuation of the virus and nothing to do with ACE2 or TMPRSS2?    2) A group of HKU scientists actually tested lung and bronchial tissues separately: https://www.nature.com/articles/s41586-022-04479-6 I have been comparing the results in this this paper with  the HKU lung tissue data. especially with respect to the various SARS-CoV-2 variants. I am able to see the remarkable similarity between the two even though the HKU folks  did it using tissues, not cells as the authors have used. The HKU folks also did not mentioned any difficulties infecting the lung tissues, which  could further my suspicion that my suggestion in (1) may be correct-- the  virus used may have been attenuated without the authors knowing.   3) The possible mechanism of SARS-CoV-2 attenuation via intrinsic disorder of  th N protein can be found at: https://pubmed.ncbi.nlm.nih.gov/35625559/ The paper basically says that attenuation can  be seen particularly with omicron with  its lower N protein disorder. The correlation between N Disorder and Viral Titer can be seen across the various stages of infection, while the HKU group reported that mutation at D614G of the S protein causes greater viral load only at initial stages of the infection. This may be a hint on what to look for if/when  the authors would like to further test their basic hypothesis as the data suggest that mutation at the S and the N affects viral infection in different ways.   4) While the paper has shown that the virus enters the cell more easily with greater abundance of ACE2 and TMPRSS27. it does not mean that attenuation of the virus did not occur in the first place.               

Round 2

Reviewer 2 Report

Improvements made